# Clinical Use of Direct Oral Anticoagulants and Reversal: Consideration for Vascular Surgeons

Alan Houben [1,*] , Vincent Bonhomme [1,2] and Marc Senard [1]

1   Department of Anesthesia and Intensive Care Medicine, Liege University Hospital, 4000 Liege, Belgium; vincent.bonhomme@chuliege.be (V.B.); msenard@chuliege.be (M.S.)
2   Anesthesia and Perioperative Neuroscience Laboratory, GIGA-Consciousness Thematic Unit, GIGA-Research, Liege University, 4000 Liege, Belgium
*   Correspondence: ahouben@chuliege.be

**Abstract:** Since their first approval in 2010, direct oral anticoagulants (DOACs) have become attractive for anticoagulant treatment. DOACs are indicated for the prevention and treatment of several cardiovascular conditions and have now emerged as leading therapeutic options. Every year, large number of patients receiving DOACs routinely are scheduled for invasive surgical procedures and need specific perioperative management. Moreover, recently published trials have provided arguments for a larger future use of DOACs, including during the postoperative period after vascular surgery and for high-risk cardiovascular patients. In this communication, we discuss the perioperative management of DOACs for patients undergoing vascular surgery.

**Keywords:** direct oral anticoagulant; vascular surgery; perioperative medicine

## 1. Introduction

Since their first approval in 2010, direct oral anticoagulants (DOACs) have become attractive alternatives to the old anticoagulation standard of care, namely the vitamin K antagonist. Compared to that, DOACs are characterized by a more immediate effect onset and offset as well as fewer pharmacological interactions with other medications or food [1]. DOACs are indicated for the prevention and treatment of several cardiovascular conditions and have now emerged as leading therapeutic options. They provide satisfaction to both clinicians and patients because of the possibility of an oral intake and the absence of a need for effect monitoring in usual clinical situations. However, a follow-up exam of kidney function is still recommended with regular monitoring of creatinine clearance. Every year, 10% of patients receiving DOACs routinely are scheduled for invasive surgical procedures and need specific perioperative management. Moreover, recently published trials have provided arguments for a larger future use of DOACs, including during the postoperative period after vascular surgery and for high-risk cardiovascular patients [2,3].

## 2. Current Indications of DOACs

The main current indications of DOACs are the prevention of stroke in non-valvular atrial fibrillation (NVAF), the treatment of deep venous thrombosis (DVT) and pulmonary embolism (PE), the prevention of recurrent DVT and PE, the prevention of DVT after hip or knee arthroplasty, the prevention of athero-thrombotic events in patients with a symptomatic peripheral artery disease, and the prevention of DVT and PE in adults hospitalized for an acute medical illness. In the near future, an expansion of indications is likely, such as for the prevention and treatment of cancer-associated DVT, the prevention of thromboembolism after percutaneous coronary intervention with NVAF, and the treatment of heparin-induced thrombocytopenia.

## 3. Pharmacology of DOACs

The main pharmacokinetic characteristics of DOACs are summarized in Table 1 [1,4,5]. The renal elimination ratio differs from one drug to the other, with a renal clearance of 80% for dabigatran. As a consequence, the monitoring of renal function is strongly recommended when these medications are used [6]. Indeed, the use of DOACs is safe in cases of moderate chronic renal insufficiency. All DOACs remain contraindicated in end-stage renal disease. Dabigatran is contraindicated in patients with severe renal impairment (stage 4, creatinine clearance between 15 mL/min and 29 mL/min). Concerning patients with hepatic impairment, the risk of bleeding complications and thrombotic events is increased in this population. No DOAC is indicated in severe hepatic impairment (Child–Pugh C). In case of moderate hepatic impairment (Child–Pugh B), dabigatran, apixaban, and edoxaban can be used without dose adjustment [1].

**Table 1.** Summary of the pharmacokinetic characteristics of DOACs [1,4,5].

| | Dabigatran | Rivaroxaban | Apixaban | Edoxaban | Betrixaban |
|---|---|---|---|---|---|
| Mechanism of action | IIa inhibitor | Xa inhibitor | Xa inhibitor | Xa inhibitor | Xa inhibitor |
| Prodrug | Yes | No | No | No | No |
| Bioavailability after oral intake (%) | 6.5 | 70 (without food) 100 (with food) | 50 | 62 | 35 |
| Tmax (h) | 0.5–2 | 2–4 | 3–4 | 0.5–2 | 3–4 |
| Half-life (h) | 12–14 | 7–9 (adults), 12 (elderly subjects > 75 years) | 12 | 10 | 19–27 |
| Plasma binding protein (%) | 35 | >90 | 87 | 55 | 60 |
| Elimination | 80 % renal, 20% hepatobiliary | 33% renal, 66% liver | 25% renal, 75% hepatobiliary | 50% renal, 50% hepatobiliary | 15% renal, 85% hepatobiliary |
| Drug to drug interaction | P-gp | P-gp, CYP3A4 | P-gp, CYP3A4 | P-gp, CYP3A4 | P-gp |
| Food to drug interaction | Prolongs Tmax to 2 h (intake with food is discouraged) | Intake with food is mandatory (especially for the 15 and 20 mg doses) | No effect | No effect | Intake with food is mandatory |
| Contraindication in case of hepatic impairment | Child–Pugh C | Child–Pugh B or C | Child–Pugh C | Child–Pugh C | Child–Pugh B or C |
| Contraindication in case of renal insufficiency | CrCl < 30 mL/min | CrCl < 15 mL/min | CrCl < 15 mL/min | CrCl < 15 mL/min | CrCl < 15 mL/min |

Footnote: Tmax = time to peak effect; half-life = plasma elimination half-life; P-gp = P glycoprotein; CYP = cytochrome P450; CrCl = creatinine clearance.

## 4. Prescription Protocols

DOAC posology varies according to the considered clinical situation and to patient-related characteristics, including age, body weight, and creatinine clearance. This results in a multiplicity of possible prescription protocols for the same drug, with a high, middle, or low range for the prescribed doses [1]. Indications and doses of DOACs are summarized in Table 2 [5,7,8].

**Table 2.** Doses and main indications of DOACs [5,7,8].

| | Dabigatran | Rivaroxaban | Apixaban | Edoxaban | Betrixaban |
|---|---|---|---|---|---|
| Prevention of DVT after hip or knee replacement | 220 mg OID, 150 mg daily if 1 | 10 mg OID | 2.5 mg BID | NA | NA |
| Prevention of stroke in NVAF | 150 mg BID, 110 mg BID if 1 | 20 mg OID, 15 mg daily if 2 | 5 mg BID, 2.5 mg BID if 3 | 60 mg OID, 30 mg OID if 4 | NA |
| Treatment of DVT and PE | 150 mg BID, 110 mg BID if 1 | 15 mg BID × 21 days then 20 mg OID or then 15 mg OID if 2 | 10 mg BID × 5 days then 5 mg BID | 60 mg OID, 30 mg OID if 4 | NA |
| Prevention of recurrent DVT and PE | 150 mg BID, 110 mg BID if 1 | 10 mg or 20 mg OID; if 2 consider 15 mg OID instead of 20 mg | 2.5 mg BID | NA | NA |
| Prevention of atherothrombotic events in symptomatic PAD | NA | 2.5 mg BID + Aspirin | NA | NA | NA |
| Prevention of stroke post-PCI with concomitant NVAF | 150 mg BID, 110 mg BID if 1 + Clopidogrel or Ticagrelor | 15 mg OID, 10 mg daily if 2 + Clopidogrel | 5 mg BID, 2.5 mg BID if 3 + Clopidogrel | 60 mg OID, 30 mg OID if 4 + Clopidogrel | NA |
| Prevention of DVT and PE in adults hospitalized for an acute medical illness | NA | NA | NA | NA | initial single dose of 160 mg, followed by 80 mg OID; initial single dose of 80 mg, followed by 40 mg OID if 5 |

Footnote: DVT = deep venous thrombosis; OID = once in a day; BID = two times in a day; NVAF = non-valvular atrial fibrillation; PAD = peripheral artery disease; 1 = creatinine clearance between 30 and 50 mL/min or age > 75 years or concomitant use of verapamil, amiodarone or quinidine; 2 = creatinine clearance between 15 and 50 mL/min; 3 = creatinine clearance between 15 and 29 mL/min or if two or three of the following criteria are met: age > 80 years, body weight ≤ 60 Kg, creatinine > 133 micromol/L; 4 = creatinine clearance between 15 and 50 mL/min or body weight ≤ 60 Kg or concomitant use of ciclosporin, dronedarone, erythromycin or ketoconazole; 5 = creatinine clearance between 15 and 29 mL/min; NA = Not applicable.

## 5. Future Expansion of Indications

The use of DOACs appears to provide benefit for high-risk cardiovascular patients. Three major trials concerning this specific population were recently published. Their results may lead to an expansion of DOACs' indications soon. First, the COMPASS trial [2] was a prospective randomized trial including more than 27,000 patients suffering from a stable atherosclerotic vascular disease. In that study, patients were allocated to one of three groups: one with a low dose of rivaroxaban (2.5 mg BID) and aspirin (100 mg OID), one with rivaroxaban (5 mg BID) alone, and one with aspirin (100 mg OID) alone. The authors demonstrated that the risk of cardiovascular events was lower in the "low dose rivaroxaban and aspirin" group than in the "aspirin alone" group, but the risk of bleeding was higher. Second, the MANAGE trial [3] was a randomized placebo-controlled trial including more than 1700 patients presenting an isolated troponin increase after non-cardiac surgery (myocardial injury after non-cardiac surgery, or MINS). They were allocated to one of two groups to receive either a mild dose of dabigatran (110 mg BID) or a placebo for two years. The authors observed a lower risk of major cardiovascular adverse events in the dabigatran group, without any significant increase in the incidence of critical bleeding complications. Third, the VOYAGER PAD trial [9] was a prospective randomized trial, which included more than 6000 patients undergoing lower limb revascularization surgery.

They were assigned to receive rivaroxaban (2.5 mg BID) and aspirin (100 mg OID), or a placebo and aspirin (100 mg OID). The authors observed a significant reduction in the incidence of cardiac or limb ischemic events in the rivaroxaban group, but a significantly higher incidence of major bleeding events. Following those studies, it might be that DOACs will now be preemptively prescribed after vascular surgery and in high-risk cardiovascular patients.

## 6. DOACs during the Perioperative Period of Vascular Surgery

Before elective vascular surgery, the discontinuation of anticoagulant treatment is obviously a rule. Indeed, vascular surgery is always considered as an intermediate- or high-bleeding-risk surgery. In this context, and to determine the optimal duration of DOACs' intake cessation, A. Godier et al. showed that, after a discontinuation for 49 to 72 h, 95% of patients had DOAC plasma concentrations of ≤30 ng/mL (Figure 1) [10], which is the threshold proposed by the French Working Group on Perioperative Hemostasis (GIHP) for performing high-bleeding-risk surgery [11].

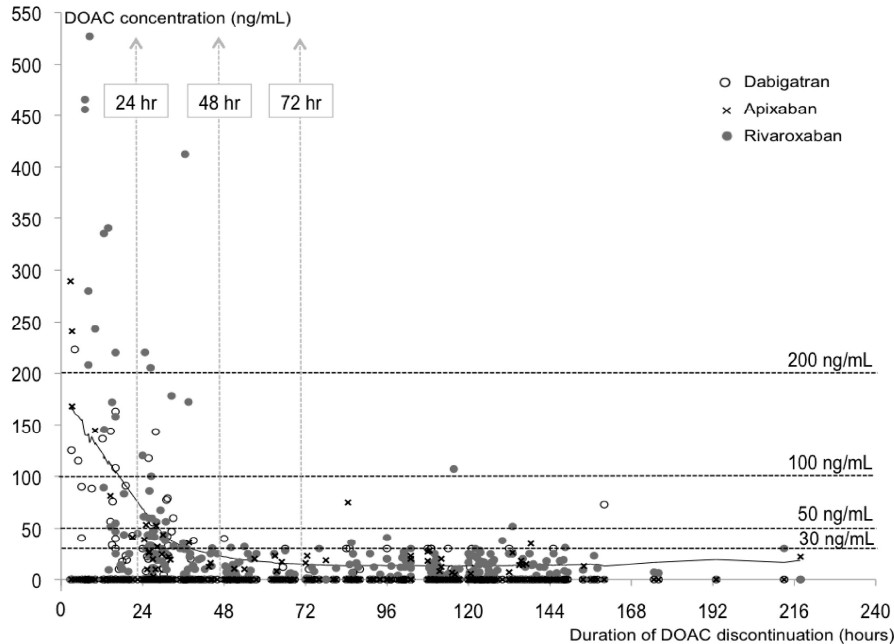

**Figure 1.** Measured DOAC concentrations as a function of the duration of DOAC intake cessation. Reproduced with permission from A. Godier [10].

Currently, most European scientific societies do not recommend bridging with low-molecular-weight heparin (LMWH) before elective surgery after DOAC cessation. Usual recommendations for cessation before surgery are 72 h for rivaroxaban, apixaban, and edoxaban, and 96 h for dabigatran and betrixaban [1,12]. Similarly, in the PAUSE study [13] and before a high-bleeding-risk surgery, DOACs were omitted for two days in patients with an ongoing dabigatran treatment and a creatinine clearance >50 mL/min or ongoing rivaroxaban or apixaban, and for four days in patients on dabigatran with a creatinine clearance ≤50 mL/min. In all patients, no heparin bridging was prescribed. This preoperative management was associated with low rates of major bleeding and thromboembolic events.

IIdarucizumab is a specific antagonist of dabigatran, providing an effective and fast reversal, but at a high cost [14]. Concerning the xabans, a specific antagonist, namely andexanet-α, obtained commercial authorization in Europe in 2019. It is currently not available in some European countries because of its prohibitive cost and some potential prothrombotic risks [15]. Before an emergency surgical procedure with a moderate to high bleeding risk in patients with an ongoing dabigatran treatment, the measurement of dabigatran plasma concentration is required, using the diluted thrombin time test. When possible,

waiting for one or two elimination half-lives before performing the procedure is the most cost-effective method to apply. If surgery cannot be postponed and plasma concentration is >50 ng/mL, it is recommended to proceed to the vascular surgery and use an infusion of idarucizumab when available [14]. In patients on xabans, a preoperative determination of the specific anti-Xa activity is recommended. The use of heparin Xa activity is also proposed to quantify apixaban and rivaroxaban's level of impregnation [16]. Delaying surgery for 12 or more hours significantly reduces the risk of bleeding. If a delay is not possible, the use of prothrombin complex concentrates before the procedure is recommended, although the efficacy of this treatment at preventing bleeding is difficult to predict [17]. The use of andexanet before a vascular surgery requiring the administration of unfractionated heparin (UFH) remains controversial. Indeed, andexanet partially neutralizes UFH and induces heparin resistance [18,19]. After potentially hemorrhagic vascular surgery, therapeutic anticoagulation may be delayed until 48–72 h after surgery because the hemorrhagic risk of resuming full-dose anticoagulation exceeds the thromboembolic risk. Postoperative thromboprophylaxis with LMWH is used until full-dose DOAC anticoagulation is resumed. Therapeutic-dose heparin anticoagulation may be used instead of DOACs if oral therapy is not possible [20].

## 7. DOACs and Locoregional Anesthesia

Before any neuraxial anesthesia such as epidural or spinal anesthesia, the GIHP recommends a 5-day DOAC discontinuation [12]. Recently published European guidelines are more liberal, proposing a 72 h discontinuation of DOACs before a neuraxial procedure, and a shorter-length cessation before superficial nerve blocks [21].

## 8. Conclusions

The number of patients receiving DOACs is increasing, with a progressive expansion of DOACs' indications, notably in high-vascular-risk patients. Before elective vascular surgery, a 72 h discontinuation of rivaroxaban, apixaban and edoxaban is recommended, while a 96 h discontinuation is necessary for dabigatran. The evidence remains limited for the management of DOACs in the context of emergency surgery. In any case, and when possible, waiting is blood-saving and possibly life-saving.

**Author Contributions:** Writing—original draft preparation, A.H. and M.S.; writing—review and editing, A.H., M.S. and V.B.; supervision, M.S. All authors have read and agreed to the published version of the manuscript.

**Funding:** This communication received no external funding.

**Institutional Review Board Statement:** Not applicable.

**Informed Consent Statement:** Not applicable.

**Data Availability Statement:** Not applicable.

**Conflicts of Interest:** The authors declare that there is no conflict of interest.

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
