# Peer review of "Clinical Use of Direct Oral Anticoagulants and Reversal: Consideration for Vascular Surgeons"

_2813-2475, doi:10.3390/jvd2020017_

Round 1
Reviewer 1 Report
Houben et al. comprehensively discussed the perioperative management of direct oral anticoagulants (DOACs) for patients undergoing vascular surgery. Their paper covered the current and future indications of DOACs, their pharmacology, recommended prescription protocols, and strategies for perioperative cessation and management in emergency situations. This communication serves to enhance the readers' understanding of the field and offer valuable guidance for clinical practice. Nonetheless, certain aspects of the paper could be further refined.
1. In this section on "Future Enlargement of Indications," it would be beneficial if the authors could provide summary and commentary language that would enhance the readers’ understanding of the three clinical studies. Rather than solely describing the studies, the authors could provide additional insights and analysis to enlighten the reader.
2. In this section on "DOACs during the perioperative period of vascular surgery," the authors reference the GIHP guidelines and the PAUSE study. However, these sources are not directly relevant to vascular surgery, which is the main focus of the article. To distinguish this review from others and provide more comprehensive information, the authors should expand on the recommendations for perioperative anticoagulant use in vascular surgery. This could include categorizing patients based on their underlying medical conditions and the type of surgery they are undergoing, and making informed decisions based on their individual thromboembolic and bleeding risks. By doing so, the authors could highlight the unique aspects of this review and provide valuable insights for clinicians in this field.
3. It is recommended that the authors consider adding a discussion on the optimal timing of resuming anticoagulation after surgery.
4. If the manuscript becomes too long, it may be appropriate to minimize the inclusion of fundamental information regarding DOACs and instead concentrate on perioperative medication strategies. This approach can make the paper more concise and focused on the core topic.
Reviewer 2 Report
Overall, the information presented in this article represents valuable information and it is well-written. It can be used as a reference for clinicians that benefit patients before undergoing vascular surgery. Having said that, the manuscript requires minor changes before it is published. They are mentioned below.
Table. 1: The authors must change the half-life values for Edoxaban to 0.5- 2
Table 1 & 2: Betrixaban is another FDA-approved drug. The authors have to give a short note on the same.
Comorbidities may alter the use of DOACs as it affects pharmacokinetics. Some DOACs cannot be used if the patient has end-stage renal disease. Therefore, the authors must add a note on renal insufficiency and hepatic impairment due to DOACs. The author could also include the Child-Pugh score if it is available to support their explanation of hepatic impairment.
In addition, the manuscript can be reviewed for minor spelling and grammar mistakes.
Round 2
Reviewer 1 Report
After undergoing revisions by the author, the article has been improved, and therefore, it is acceptable for publication.